# MEMSEARCHER: TRAINING LLMs TO REASON, SEARCH AND MANAGE MEMORY VIA END-TO-END REINFORCEMENT LEARNING

## ABSTRACT

Traditional search agents concatenate the entire interaction history into the LLM context, preserving information integrity but producing long, noisy contexts, resulting in high computation and memory costs. In contrast, using only the current turn avoids this overhead but discards essential information. This trade-off limits the scalability of search agents. To address this challenge, we propose MemSearcher, an agent workflow that iteratively maintains a compact memory and combines the current turn with it. At each turn, MemSearcher fuses the user's question with the memory to generate reasoning traces, perform search actions, and update memory to retain only information essential for solving the task. This design stabilizes context length across multi-turn interactions, improving efficiency without sacrificing accuracy. To optimize this workflow, we introduce multi-context GRPO, an end-to-end RL framework that jointly optimize reasoning, search strategies, and memory management of MemSearcher Agents. Specifically, multi-context GRPO samples groups of trajectories under different contexts and propagates trajectory-level advantages across all conversations within them. Trained on the same dataset as Search-R1, MemSearcher achieves significant improvements over strong baselines on seven public benchmarks: +11% on Qwen2.5-3B-Instruct and +12% on Qwen2.5-7B-Instruct relative average gains. Notably, the 3B-based MemSearcher even outperforms 7B-based baselines, demonstrating that striking a balance between information integrity and efficiency yields both higher accuracy and lower computational overhead. Our code and models will be publicly available.

## 1 INTRODUCTION

Large Language Models (LLMs) (Team, 2024; Achiam et al., 2023) have demonstrated impressive performance in understanding and generating natural language, as well as in solving complex tasks in the real world. Despite their strengths, LLMs still exhibit notable shortcomings in addressing knowledge-acquisition tasks (Wei et al., 2024; He et al., 2024). These shortcomings arise from their insufficient long-tailed and up-to-date knowledge in specific domains and susceptibility to hallucinations (Xu et al., 2024; Zhang et al., 2025c).

A promising strategy to mitigate these issues is to integrate search engines with LLMs, allowing them to access external and up-to-date information. Considerable efforts have been devoted to this area in recent years. In Retrieval-Augmented Generation (RAG) methods (Gao et al., 2023; Zhao et al., 2024), a search engine is used to select relevant documents according to the the input of the LLM, and the retrieved documents are fed into the LLM context to generate the final response. While straightforward, these methods often rely on predefined pipelines (Zhou et al., 2025a; Zhu et al., 2025b) and do not fully explore the potential of LLMs in leveraging search engines. To address this limitation, search agents, which treat a search engine as a tool, have been developed.

A representative paradigm to build search agents is ReAct (Yao et al., 2023). In ReAct, the interactions between the agent and the search engine are modeled as a multi-turn conversation, which means that the entire interaction history is incorporated into the context of the agent's backbone LLM. This paradigm provides the agent with fine-grained information—including all reasoning pro-

cesses, performed actions and corresponding tool responses from previous interactions—to support more effective decision-making. However, the continuously appended interaction history leads to unbounded growth of the context of the LLM, which substantially increases the GPU memory and computational overhead.

In this paper, we introduce MemSearcher, an agentic workflow that maintains a compact, iteratively updated memory throughout interactions, preserving only the information deemed essential for addressing the user's question. At each turn, MemSearcher provides the backbone LLM with two succinct inputs, the user question and a compact memory, rather than the entire, ever-growing interaction history. The LLM first generates the reasoning trace and performs an action based on it. After the new observation is returned to the agent by the environment, the LLM then functions as a memory manager to update the memory based on the previous memory and the current interaction. Since the number of tokens in the memory is restricted by a predefined maximum length, this design keeps per-turn contexts short and stable while preserving salient facts and intermediate findings across multi-turn reasoning and interactions.

Since current LLMs have not been optimized under the MemSearcher workflow, they are not yet capable of mastering it. We employ Reinforcement Learning (RL) (Wiering & Van Otterlo, 2012) to train MemSearcher agents, which enables models to improve by leveraging their self-generated samples as optimization targets. Among RL algorithms, Group Relative Policy Optimization (GRPO) (Shao et al., 2024) has recently emerged as the most widely adopted method, as it improves LLM abilities while optimizing the GPU memory usage of Proximal Policy Optimization (PPO) (Schulman et al., 2017). We extend vanilla GRPO to multi-context GRPO to facilitate the training of MemSearcher agents, whose trajectories consist of multiple conversations under different contexts. Specifically, multi-context GRPO propagates trajectory-level advantages to each conversation among them and subsequently treats every conversation as an independent optimization target. This extension enables a stable and scalable training for MemSearcher-based agents.

We use the same data as Search-R1 (Jin et al., 2025) to train MemSearcher from scratch on Qwen2.5-3B-Instruct and Qwen2.5-7B-Instruct, and conduct extensive evaluation on a range of public knowledge-acquisition benchmarks that need reasoning and external information retrieval. Our MemSearcher agents demonstrate significant improvement over the baseline methods, yielding 11% and 12% increases on different models, respectively. Moreover, MemSearcher also achieves higher efficiency than the ReAct paradigm. Compared to ReAct-based search agents, which exhibits a steady increase in token numbers with interaction turns, our MemSearcher agents maintain nearly constant token counts within contexts.

We summarize our main contributions as follows.

- We introduce MemSearcher, an agentic workflow that leverages the backbone LLM as a memory manager to iteratively maintain a compact memory, preserving only the essential information necessary for answering the user's question and thereby eliminating the need to append the entire interaction history to the LLM context.

- We develop search agents based on MemSearcher, and utilize multi-context GRPO, a natural extension of GRPO, to optimize LLMs to reason, leverage search engines and manage memory simultaneously. Multi-context GRPO provides end-to-end RL training for trajectories that contain multiple conversations under different contexts.

- We use the same data as Search-R1 to train our search agents. The evaluation on seven public benchmarks demonstrates the effectiveness and efficiency of our method, with two LLMs achieving average relative improvements of 11% and 12%, respectively. Compared with ReAct-based search agents, which exhibit a nearly linear increase in token numbers during interactions, MemSearcher agents maintain lower and more stable token counts.

## 2 BACKGROUND

### 2.1 PRELIMINARY: REACT

ReAct (Yao et al., 2023), which integrates reasoning and acting, has become the most popular paradigm for building LLM-based agents Jin et al. (2025); Chen et al. (2025).

Figure 1: Comparison between ReAct (Top) and MemSearcher (Bottom). The dashed box illustrates the content included in the LLM context. While ReAct continuously appends all thoughts, actions and observations into the LLM context, MemSearcher iteratively updates a compact memory that retains only the essential information.

As shown in Figure 1 (Top), the core idea of ReAct is straightforward: a ReAct trajectory is a multi-turn conversation, and each turn is an interaction between the LLM agent and the environment, containing thought, action, and observation. At each turn, the LLM first generates a thought within `<think> </think>`, and then performs an action in `<tool_call> </tool_call>`, to interact with the environment, after which the environment provides an observation between `<tool_response> </tool_response>` in response to the performed action.

Specifically, we assume that at the $i$-th turn, the agent generates a thought $t_i$, takes an action $a_i$, and receives an observation $o_i$. In particular, $o_0 = q$ represents the observation prior to the first turn, where $q$ denotes the user's question. Then, the context to the LLM is as follows:

$$c_i = (q, t_1, a_1, o_1, \cdots, t_{i-1}, a_{i-1}, o_{i-1}). \tag{1}$$

At the $i$-th turn, the agent generates the thought $t_i$ and performs the corresponding action $a_i$, following policy $\pi(t_i, a_i | c_i)$.

## 2.2 LIMITATIONS OF REACT

Although straightforward and simple, this paradigm leads to a continuous increase in the number of tokens in the LLM context, due to its design of appending all previous thoughts, actions and observations. This increase is almost linear with the number of interaction turns, placing significant pressure on the inference of LLMs. For example, Liu et al. (2023) find that LLMs do not reliably make use of information from long contexts. Hsieh et al. (2024) demonstrate that LLMs exhibit large performance drops as the context length increases. Wu et al. (2024) reveal that LLMs show a significant accuracy drop on memorizing information across sustained multi-turn interactions. In addition, in the context of search agents, the observations are passages retrieved by the search engine, which often include substantial noise and information irrelevant to answering the user's question. This further constrains the performance and scalability of ReAct-based search agents. Moreover, the linear growth in the number of tokens leads to increased memory consumption and computational overhead. Since the computational complexity of LLMs scales as $O(n^2)$ with the number of tokens $n$, the computational cost of these search agents increases quadratically with the number of interaction turns. Consequently, more efficient and scalable approaches for building search agents need to be explored.

## 3 METHOD

### 3.1 OVERVIEW OF MEMSEARCHER

The MemSearcher workflow is illustrated in Figure 1 (Bottom). At the $i$-th turn, the LLM receives only two inputs: the user's question $q$, enclosed within `<question> </question>` tags, and a

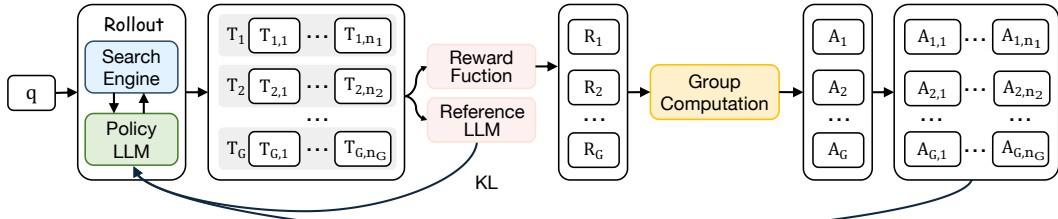

Figure 2: Multi-context GRPO. In rollout, we sample a group of trajectories $\{T_i\}_{i=1}^{G}$ for question $q$. The $i$-th trajectory $T_i$ consists of multiple conversations $\{T_{i,j}\}_{j=1}^{n_i}$ under different contexts. Then, we compute rewards $\{R_i\}_{i=1}^{G}$, and derive the trajectory-level advantages $\{A_i\}_{i=1}^{G}$ from these rewards. We propagate trajectory-level advantages to each conversation within them, *i.e.* $A_{i,j} = A_i$, and treat each conversation as an independent optimization target to update the policy LLM.

compact memory $m_{i-1}$ expressed in natural language, between `<memory> </memory>`, which encapsulates all the relevant information considered helpful to answer the question so far. In particular, the memory $m_0$ prior to the first turn is empty. Therefore, the context to the LLM at the $i$-th is formulated as:

$$c_i = (q, m_{i-1}). \tag{2}$$

After reading the user's question and the previous memory, the LLM generates a thought $t_i$ enclosed within `<think> </think>` and performs an action $a_i$ between `<tool_call> </tool_call>` tags, following policy $\pi(t_i, a_i | c_i)$. As the action is executed, the environment returns the observation $o_i$ within `<tool_response> </tool_response>` tags to the agent.

After receiving $o_i$, MemSearcher overwrites the previous memory to a updated one for the next turn. The LLM are asked to carefully reads $o_i$ and incorporates any new information that helps to answer the question, while preserving all relevant details from the previous memory $m_{i-1}$. The resulting memory are denoted as $m_i$.

Different from ReAct, which continuously concatenates all historical thoughts, actions and observations into the LLM context, MemSearcher compresses only the essential information into a compact memory. Since the number of tokens in the memory never exceeds a predefined maximum length, MemSearcher maintains the context within a few thousands of tokens while retaining important information through iterative updates of the memory. This process continues iteratively, until the maximum number of interactions is reached or sufficient information is gathered and the LLM generates a final answer as its action.

Specifically, under the setting of search agents designed to solve knowledge-acquisition tasks by leveraging search engines as tools, action $a_i$ takes one of the following two forms: (1) providing a final answer in `\boxed{}` to the user's question and terminating the interactions, (2) issuing a search engine call with a query to obtain additional information to answer the question. If the latter is chosen, the observation $o_i$ is the relevant passages retrieved from search engines in response to the search query.

## 3.2 RL TRAINING ALGORITHM

In this subsection, we introduce multi-context GRPO, the training algorithm of our MemSearcher agents. Figure 2 illustrates the overview of multi-context GRPO.

We use end-to-end reinforcement learning (RL) to train our MemSearcher agents, since it allows models to evolve themselves through their self-generated samples. In contrast, Supervised Fine-Tuning (SFT) requires costly, carefully curated high-quality trajectories, such as Li et al. (2025a); Sun et al. (2025b); Wu et al. (2025); Schick et al. (2023). For RL, we utilize Group Relative Policy Optimization (GRPO) (Shao et al., 2024), as it optimizes the memory usage of Proximal Policy Optimization (PPO) (Schulman et al., 2017) and has recently become the most widely adopted RL algorithm for RLVR due to its effectiveness (Guo et al., 2025).

Vanilla GRPO samples a group of trajectories $\{T_1, T_2, \cdots, T_G\}$ for each question $q$, and then optimizes the policy model $\pi_\theta$ by maximizing the following objective:

$$\mathcal{J}_{GRPO}(\theta) = \mathbb{E}[q \sim D, \{T_i\}_{i=1}^{G} \sim \pi_{\theta_{old}}(\cdot|q)]$$

$$\frac{1}{G} \sum_{i=1}^{G} \left( \min\left(r_i(\theta)A_i, \text{clip}\left(r_i(\theta), 1 - \epsilon, 1 + \epsilon\right) A_i\right) - \beta \mathbb{D}_{KL}\left(\pi_\theta || \pi_{ref}\right) \right), \quad (3)$$

where

$$r_i(\theta) = \frac{\pi_\theta(T_i|q)}{\pi_{\theta_{old}}(T_i|q)}. \quad (4)$$

$A_i$ represents the normalized advantage, calculated by using the rewards $\{R_1, R_2, \cdots, R_G\}$ within each group:

$$A_i = \frac{R_i - \text{mean}(\{R_1, R_2, \cdots, R_G\})}{\text{std}(\{R_1, R_2, \cdots, R_G\})}. \quad (5)$$

In the training of MemSearcher, each trajectory consists of multiple conversations under different LLM contexts. Therefore, we extend the vanilla GRPO algorithm to a natural extension, multi-context GRPO, as illustrated in Figure 2. Specifically, we assume that trajectory $T_i$ contains $n_i$ conversations, represented as $\{T_{i,1}, T_{i,2}, \cdots, T_{i,n_i}\}$. According to Section 3.1, the $j$-th conversation can be represented as:

$$T_{i,j} = \begin{cases} (q, m_{i,j-1}, t_{i,j}, a_{i,j}, o_{i,j}, m_{i,j}), & \text{if } j = 1, 2, \cdots, n_i - 1 \\ (q, m_{i,j-1}, t_{i,j}, a_{i,j}), & \text{if } j = n_i \end{cases} \quad (6)$$

where memories $m_{i,j-1}$ and $m_{i,j}$, thought $t_{i,j}$ and action $a_{i,j}$ are generated by the policy model, and observation $o_{i,j}$ is the retrieved text from the search engine.

We compute reward $R_i$ for each trajectory, and calculate its advantage $A_i$ within the group using Equation 5. Then, we uniformly propagate this advantage to all conversations whithin the trajectory, and use each conversation as an independent target to optimize the policy model. The training objective is formulated as:

$$\mathcal{J}_{GRPO}(\theta) = \mathbb{E}[q \sim D, \{T_{i,j}\}_{i=1}^{G} \sim \pi_{\theta_{old}}(\cdot|q, m_{i,j-1})]$$

$$\frac{1}{\sum_{i=1}^{G} n_i} \sum_{i=1}^{G} \sum_{j=1}^{n_i} \left( \min\left(r_{i,j}(\theta)A_{i,j}, \text{clip}\left(r_{i,j}(\theta), 1 - \epsilon, 1 + \epsilon\right) A_{i,j}\right) - \beta \mathbb{D}_{KL}\left(\pi_\theta || \pi_{ref}\right) \right),$$

$$(7)$$

where

$$r_{i,j}(\theta) = \frac{\pi_\theta(T_{i,j}|q, m_{i,j-1})}{\pi_{\theta_{old}}(T_{i,j}|q, m_{i,j-1})} \text{ and } A_{i,j} = A_i. \quad (8)$$

Notably, conversation $t_{i,j}$ consists of tokens from both the policy model and the search engine. Following previous RL-based search agents, such as Search-R1 (Jin et al., 2025) and ReSearch (Chen et al., 2025), we use loss masking for the tokens from the search engine, ensuring the policy gradient objective is computed only over model-generated tokens and thereby stabilizing RL training.

## 3.3 REWARD MODELING

The reward serves as the primary training signal in RL, guiding the optimization process of models. During the training of MemSearcher, we only adopt a simple reward function on the generated samples. Similar to DeepSeek-R1 (Guo et al., 2025), our reward function considers two parts: format reward and answer reward.

- **Format Reward**: It checks whether the rollout correctly follows our predefined format, including the correctness of usage of tags and the existence of \boxed{} in the answer.
- **Answer Reward**: A rule-based reward assesses the correctness of the model's response. It is calculated by using the F1 score between the final answer inside \boxed{} and the ground truth.

Table 1: Performance comparison. Exact Match (EM) is used as the evaluation metric. The best performance is highlighted in **bold**, while the second-best performance is indicated with an underline. Among these methods, R1-Searcher and ZeroSearch interact with the realistic web environment during their evaluation, while other methods, including MemSearcher, interact only with local knowledge base. MemSearcher based on Qwen2.5-3B-Instruct achieves a higher average score than other methods based on Qwen2.5-7B-Instruct.

| Methods | NQ | TriviaQA | PopQA | HotpotQA | 2wiki | Musique | Bamboogle | Avg. |
|---|---|---|---|---|---|---|---|---|
| **Qwen2.5-3B-Instruct** | | | | | | | | |
| Direct Answer | 10.6 | 28.8 | 10.8 | 14.9 | 24.4 | 2.0 | 2.4 | 13.4 |
| CoT | 2.3 | 3.2 | 0.5 | 2.1 | 2.1 | 0.2 | 0.0 | 1.5 |
| IRCoT | 11.1 | 31.2 | 20.0 | 16.4 | 17.1 | 6.7 | 24.0 | 18.1 |
| RAG | 34.8 | 54.4 | 38.7 | 25.5 | 22.6 | 4.7 | 8.0 | 27.0 |
| Search-o1 | 23.8 | 47.2 | 26.2 | 22.1 | 21.8 | 5.4 | 32.0 | 25.5 |
| Search-R1 | 34.1 | 54.5 | 37.8 | 32.4 | 31.9 | 10.3 | 26.4 | 32.5 |
| ReSearch | 20.4 | 33.5 | 17.3 | 35.6 | 39.3 | 17.3 | 37.6 | 28.7 |
| AutoRefine | 43.6 | 59.7 | 44.7 | 40.4 | 38.0 | 16.9 | 33.6 | 39.6 |
| ZeroSearch | 41.4 | 57.4 | 44.8 | 27.4 | 30.0 | 9.8 | 11.1 | 31.7 |
| MemSearcher | **47.0** | **63.8** | **47.9** | **43.9** | **43.5** | **17.9** | **42.4** | **43.8** |
| **Qwen2.5-7B-Instruct** | | | | | | | | |
| Direct Answer | 13.4 | 40.8 | 14.0 | 18.3 | 25.0 | 3.1 | 12.0 | 18.1 |
| CoT | 4.8 | 18.5 | 5.4 | 9.2 | 11.1 | 2.2 | 23.2 | 10.6 |
| IRCoT | 22.4 | 47.8 | 30.1 | 13.3 | 14.9 | 7.2 | 22.4 | 23.9 |
| RAG | 34.9 | 58.5 | 39.2 | 29.9 | 23.5 | 5.8 | 20.8 | 30.4 |
| Search-o1 | 15.1 | 44.3 | 13.1 | 18.7 | 17.6 | 5.8 | 29.6 | 20.6 |
| Search-R1 | 39.3 | 61.0 | 39.7 | 37.0 | 41.4 | 14.6 | 36.8 | 38.5 |
| ReSearch | 40.9 | 63.7 | 44.6 | 43.5 | 47.6 | 22.3 | 42.4 | 43.6 |
| R1-Searcher | 40.4 | 52.2 | 41.0 | 44.2 | **51.3** | 15.8 | 36.8 | 40.2 |
| ZeroSearch | 43.6 | 65.2 | **48.8** | 34.6 | 35.2 | 18.4 | 27.8 | 39.1 |
| MemSearcher | **52.7** | **68.1** | 47.8 | **50.8** | 48.6 | **25.8** | **48.8** | **48.9** |

The reward function is formulated as:

$$R = \begin{cases} 0, & \text{if format is incorrect} \\ 0.1, & \text{if format is correct but F1}(a_{\text{pred}}, a_{\text{gold}}) \text{ is } 0 \\ \text{F1}(a_{\text{pred}}, a_{\text{gold}}), & \text{if format is correct and F1}(a_{\text{pred}}, a_{\text{gold}}) \text{ is not } 0 \end{cases} \quad (9)$$

where $a_{\text{pred}}$ is the final answer extracted from the model's response, $a_{\text{gold}}$ is the ground truth, and F1$(a_{\text{pred}}, a_{\text{gold}})$ is the F1 score between $a_{\text{pred}}$ and $a_{\text{gold}}$.

# 4 EXPERIMENTS

## 4.1 EXPERIMENT SETUPS

**Baselines.** We compare MemSearcher against three categories of baseline methods, including: (1) Inference without retrieval, such as Direct inference and Chain-of-Thought (CoT) reasoning (Wei et al., 2022); (2) Inference with Retrieval, such as RAG (Lewis et al., 2020), IRCoT (Trivedi et al., 2022a), and Search-o1 (Li et al., 2025b); (3) RL-based search agents, such as Search-R1 (Jin et al., 2025), ReSearch (Chen et al., 2025), AutoRefine (Shi et al., 2025), R1-Searcher (Song et al., 2025), and ZeroSearch (Sun et al., 2025a). Among these baselines, R1-Searcher and ZeroSearch interact with the realistic web environment via Google Web Search during their evaluation.

**Benchmarks and Evaluation Metrics.** We compare MemSearcher and the baseline methods on a range of public benchmarks that encompass search with reasoning challenges, such as Natural Questions (NQ) (Kwiatkowski et al., 2019), TriviaQA (Joshi et al., 2017), PopQA (Mallen et al., 2022), Bamboogle (Press et al., 2022), Musique (Trivedi et al., 2022b), HotpotQA (Yang et al., 2018), and 2WikiMultiHopQA (Ho et al., 2020). We use Exact Match (EM) as the evaluation metric, where the prediction is correct if it matches the ground truth answer exactly.

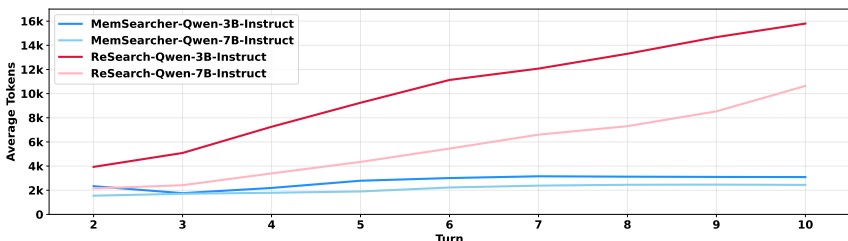

Figure 3: Comparison of the average token number in the LLM context between MemSearcher and ReAct-based ReSearch.

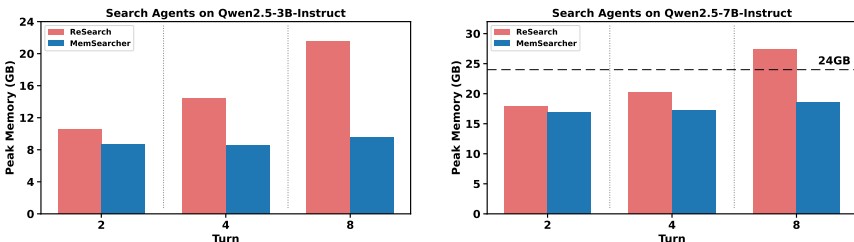

Figure 4: Peak GPU memory usage (GB) comparison between MemSearcher and ReSearch.

**Implementation Details.** We conduct our training and evaluation on Qwen2.5-3B-Instruct and Qwen2.5-7B-Instruct. We use the 2018 Wikipedia dump (Karpukhin et al., 2020) as the knowledge source and E5 (Wang et al., 2022) as the retriever. We conduct the training based on verl (Sheng et al., 2025), constrain the model to an 8K context window, and set the maximum tokens in the memory to 1,024 tokens. For training, we follow Search-R1, using its fully open training data, including the training splits of two datasets, NQ (Kwiatkowski et al., 2019) and HotpotQA (Yang et al., 2018), to form the dataset for training. Within the datasets used for training, NQ is a general question answering dataset, while HotpotQA is a multi-hop question answering dataset. For evaluation, we systematically test MemSearcher and the baseline methods on seven datasets, covering both in-domain and out-of-domain scenarios. This setup enables us to rigorously assess not only how well the models generalize to questions that resemble the training distribution, but also how robust they are when applied to domains that differ from the training data. Such a comprehensive evaluation provides deeper insights into the effectiveness of our approach under varied conditions.

## 4.2 MAIN RESULTS

In Table 1, we provide a comprehensive performance comparison between MemSearcher and the baseline methods across the evaluated benchmarks. Several key observations can be drawn from these results: (1) When trained on the same datasets as Search-R1, MemSearcher consistently outperforms the baseline methods, demonstrating the superior effectiveness of our method. These performance improvements are consistently observed across both in-distribution benchmarks such as NQ and HotpotQA, and out-of-distribution benchmarks, such as TriviaQA, PopQA, 2WikiMulti-HopQA, Musique and Bamboogle. (2) Remarkably, even when using a smaller backbone model, *i.e.* Qwen2.5-3B-Instruct, MemSearcher achieves an average EM score 43.8 on the seven benchmarks, higher than those of the baseline methods based on the lager model, *i.e.* Qwen2.5-7B-Instruct, suggesting that MemSearcher makes more effective use of model capacity. (3) Furthermore, MemSearcher surpasses the baseline methods that rely on the realistic web search engine. Specifically, MemSearcher achieves superior performance compared to R1-Searcher and ZeroSearch, both of which depend on Google Web Search to retrieve external information during their evaluation.

In addition to the improvement in performance, MemSearcher also achieves superior token efficiency compared with ReAct-based search agents, since it eliminates the need to append all historical thoughts, actions and observations into the LLM context, as discussed in Section 3. To validate this, we record the number of tokens in the LLM contexts of MemSearcher and ReAct-based ReSearch

Table 2: Comparison between models with and without training. Exact Match (EM) is used as the evaluation metric. The better performance is highlighted in **bold**.

| Methods | General QA | | | Multi-Hop QA | | | | |
|---|---|---|---|---|---|---|---|---|
| | NQ | TriviaQA | PopQA | HotpotQA | 2wiki | Musique | Bamboogle | Avg. |
| **Qwen2.5-3B-Instruct** | | | | | | | | |
| *w/o* training | 16.4 | 23.8 | 22.5 | 11.9 | 11.0 | 3.7 | 11.2 | 14.4 |
| *w/* training | **47.0** | **63.8** | **47.9** | **43.9** | **43.5** | **17.9** | **42.4** | **43.8** |
| **Qwen2.5-7B-Instruct** | | | | | | | | |
| *w/o* training | 22.1 | 41.2 | 23.5 | 27.4 | 27.8 | 11.6 | 27.2 | 25.8 |
| *w/* training | **52.7** | **68.1** | **47.8** | **50.8** | **48.6** | **25.8** | **48.8** | **48.9** |

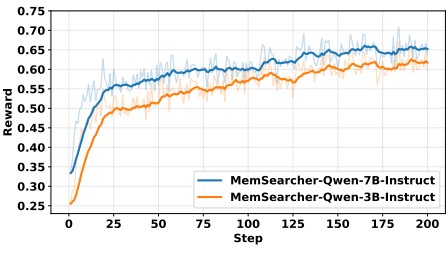 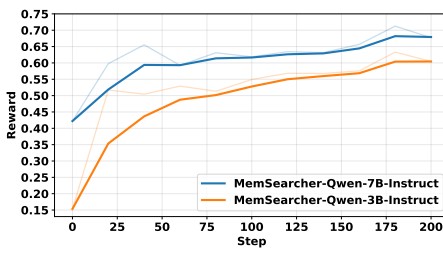

(a) Training Reward                          (b) Validation Reward

Figure 5: Training and validation reward during training. The validation is conducted on a part of development set of HotpotQA with 100 randomly selected samples, and conducted every 20 steps during training. The curves are smoothed for clarity.

at each turn and calculate their average across the evaluated datasets. The results are illustrated in Figure 3. Compared to ReSearch, which exhibits an almost linear increase in token consumption during the interaction process, MemSearcher maintains substantially lower and more stable token counts. The efficiency gain is primarily attributed to the design of MemSearcher, which iteratively updates a compact memory as context to preserve only the essential information for the question throughout the interactions. Moreover, we present the preak GPU memory usage comparison in Figure 4. We can observe that MemSearcher supports more scalable and cost-efficient multi-turn interactions in search agents.

### 4.3 FURTHER ANALYSIS

#### 4.3.1 DO WE NEED RL TRAINING?

To investigate the impact of RL training on the performance of MemSearcher, we perform a comparative analysis. The baselines are Qwen2.5-3B-Instruct and Qwen2.5-7B-Instruct models, both of which are integrated with the MemSearcher workflow but do not undergo RL training. As shown in Table 2, the models without RL training demonstrate a pronounced performance degradation across all evaluated benchmarks. This observation highlights the necessity of RL training in equipping models with the ability to effectively interact with both the search engine and memory, thereby enhancing their overall functionality and task-solving ability.

#### 4.3.2 TRAINING AND VALIDATION REWARD.

We present the curves of training and validation reward in Figure 5, which offer an intuitive view of the models' learning dynamics during training. For the validation, we construct a validation dataset by randomly sampling 100 examples from the development set of HotpotQA. We conduct validation at fixed intervals, specifically every 20 training steps. The observed reward patterns reveals the following two phases of learning: (1) Early stage (first 25 steps). In this phase, the reward increases sharply. This improvement indicates that the models rapidly acquire the fundamental ability to interact effectively with the search engine and memory. (2) Later stage (after 25 steps). In contrast, the reward grows at a more gradual pace. This improvement suggests that the models are refining their strategy, progressively enhancing its capacity to exploit the search engine and manage memory.

The difference between these two stages underscores the transition from basic skill acquisition to more advanced optimization of reasoning behaviors.

## 5 RELATED WORK

### 5.1 LARGE LANGUAGE MODELS WITH SEARCH ENGINES

Although Large Language Models (LLMs) (Team et al., 2025; Comanici et al., 2025; Zeng et al., 2025) have made significant progress in solving complex tasks in the real world (Guo et al., 2024), they often lack knowledge in specific domains (Peng et al., 2023; Li et al., 2023). To address these issues, Retrieval-Augmented Generation (RAG) integrates search engines (Xiao et al., 2024; Zhuang et al., 2024) with LLMs to provide relevant external information. In a typical RAG pipeline (Lewis et al., 2020; Yue et al., 2024; Xiong et al., 2025), a search engine first selects relevant documents based on the input query, and then the retrieved content is fed into an LLM to produce responses. Previous studies on RAG guides LLMs through processes such as search query generation and decomposition (Yu et al., 2022; Press et al., 2022). Although RAG enhances the performance of LLMs, it faces challenges related to the retrieval of irrelevant information (Zhu et al., 2025a; Jin et al., 2024a) and the absence of sufficiently useful context (Jiang et al., 2023). In addition to RAG, another approach to integrate external search engines with LLMs is to treat search engines as tools and LLMs as agents, named search agent (Zong et al., 2024). For example, ReAct (Yao et al., 2023) integrates search into the reasoning process by interleaving it with Chains-of-Thought (CoT) (Wei et al., 2022) steps. Recent studies (Jin et al., 2025; Chen et al., 2025; Zheng et al., 2025) develop agentic reinforcement learning (RL) (Zhang et al., 2025b) for search agents, based on multi-turn chat. Although effective, current RL-based search agents (Wu et al., 2025; Tao et al., 2025) primarily adhere to the ReAct workflow, lacking the exploration of more efficient paradigms.

### 5.2 CONTEXT MANAGEMENT

Most LLM agents utilize ReAct (Yao et al., 2023) for context management, which incorporates the entire interaction history between the LLM and the environment into the LLM context. While simple, it leads to prolonged token sequences and reduced efficiency. To address these issues, memory mechanisms are proposed to manage the context of LLMs. RAG-style memory systems (Jimenez Gutierrez et al., 2024; Zhong et al., 2024) treat memory as an external knowledge source, similar to that in RAG, and use predefined management strategies to store, integrate and retrieve relevant information (Zhu et al., 2023). Token-level memory systems (Jin et al., 2024b; Zhou et al., 2025b; Orlicki, 2025) equip models with explicit, trainable context managers and optimize them via SFT or RL algorithms such as PPO (Schulman et al., 2017), allowing agents to regulate their memory at the token level. For example, Wang et al. (2024) and Wang et al. (2025) maintain a fixed set of latent tokens serves as memory, and Yang et al. (2024) equip LLMs with a forget-resistant memory for evolving context. MemAgent (Yu et al., 2025) reforms long-context processing as an agent task, maintains a token-level memory alongside the LLM to compress long-context inputs into more concise, informative summaries. Structured memory systems (Zeng et al., 2024) organize and encode information in structured representation, such as knowledge graph in Zep (Rasmussen et al., 2025), the atomic memory units in A-MEM (Xu et al., 2025), and the hierarchical graph-based memory in Mem0 (Chhikara et al., 2025) and G-Memory (Zhang et al., 2025a). In this paper, we utilize the backbone LLM of search agents as a memory manager, and optimize it for reasoning, action, and memory management via end-to-end multi-context GRPO algorithm.

## 6 CONCLUSION

In this paper, we propose MemSearcher, an agentic workflow that retains a compact memory as LLM context throughout the interaction process between the agent and the environment, thereby eliminating the need to append all historical thoughts, actions and observations, as in the ReAct paradigm. We utilize a natural extension of GRPO, namely multi-context GRPO, to optimize search agents based on our workflow in an end-to-end fashion. These agents demonstrate superior performance across a range of public benchmarks compared with previous ReAct-based baselines, while maintaining nearly constant token consumption during interactions with the environment, highlighting more scalable and cost-efficient multi-turn interactions in search agents.

## 7    ETHICS STATEMENT

We affirm our strict adherence to the ICLR Code of Ethics in all aspects of this work. Our research does not involve human participants, personal data, or sensitive information. The datasets used (*e.g.*, NQ, HotpotQA, and other public QA benchmarks) are entirely publicly available and widely used within the research community, ensuring no privacy or security concerns arise.

Our experiments are focused on algorithmic innovation in context management and reinforcement learning (RL) for large language model (LLM)-based agents. While our methods aim to improve computational efficiency and accuracy, care should be taken if applying them in domains where model outputs may have real-world impact. We advise thorough assessment of fairness, reliability, and bias mitigation in any such downstream use. Finally, we intend to make our code and models publicly available to promote transparency and foster ongoing research integrity.

## 8    REPRODUCIBILITY STATEMENT

We are committed to ensuring the reproducibility of our work. All datasets used in our experiments are publicly available and described in Section 4.1 of this paper. Implementation details, including models, hyperparameters, training procedures, and evaluation metrics, are described in Section 4.1 as well as in Appendix A.2. Additionally, we will release all code and trained models.

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

# A APPENDIX

## A.1 LLM USAGE

We used OpenAI's ChatGPT to help polish the language and improve the readability of the manuscript. Specifically, ChatGPT was used for grammar checking and sentence rephrasing. We list our prompt for using OpenAI's ChatGPT to help polish writing as follows.

> **Prompt for Using OpenAI's ChatGPT to Help Polish Writing**
>
> ```
> Below is a paragraph from an academic paper. Polish the
>     writing to meet the academic style,improve the spelling,
>     grammar, clarity, concision and overall readability.
>     Furthermore, list all modification and explain the
>     reasons to do so in markdown table. \\
> Paragraph: {paragraph}
> ```

## A.2 TRAINING HYPERPARAMETERS

We train MemSearcher agents with full parameter optimization and gradient checkpointing. We show some important training hyperparameters in Table 3.

Table 3: Training details of MemSearcher.

| Parameter | Value |
|---|---|
| Learning Rate | 1e-6 |
| Train Batch Size | 256 |
| Number of Training Epochs | 1 |
| Number of Rollout | 5 |
| Rollout Temperature | 1.0 |
| KL Loss Coefficient | 0.001 |
| Clip Ratio | 0.2 |

## A.3 DETAILS OF EVALUATED DATASETS

We evaluate MemSearcher agents on the following public question answering datasets:

- **Natural Questions (NQ)** (Kwiatkowski et al., 2019), a QA dataset with questions consisting of real anonymized, aggregated queries issued to the Google search engine.
- **TriviaQA** (Joshi et al., 2017), a large scale challenging dataset with relatively complex, compositional questions, requireing more reasoning to find answers.
- **PopQA** (Mallen et al., 2022), a dataset consisting of 14k questions covering factual information in the long tail.
- **Bamboogle** (Press et al., 2022), a manually constructed dataset with multi-hop questions, where all questions are sufficiently difficult to be unanswerable by a single search engine call.
- **Musique** (Trivedi et al., 2022b), a QA dataset with 25K questions that by require proper multi-hop reasoning.
- **HotpotQA** (Yang et al., 2018), a dataset with 113k Wikipedia-based question-answer pairs, where the questions requires finding and reasoning over multiple supporting documents to answer, and and not constrained to any existing knowledge bases or knowledge schemas.
- **2WikiMultiHopQA** (Ho et al., 2020), a multi-hop QA dataset that uses both structured and unstructured data. It is challenging for multi-hop models and explicitly ensures that multi-hop reasoning is required.

## A.4 CASE STUDY

To provide a more intuitive understanding of the utility of MemSearcher, we present a case study in Table 4. This case is drawn from the evaluation of MemSearcher based on Qwen2.5-7B-Instruct. The text enclosed by `<think>` and `</think>`, `<tool_call>` and `</tool_call>`, as well as `<memory>` and `</memory>` is generated by the model. The text enclosed by `<tool_response>` and `</tool_response>` is retrieved from the search engine. This case demonstrates that the model can effectively maintain a compact memory, retaining only the essential information necessary to solve the question.

Table 4: A case study of MemSearcher.

**Question:** What was the form of the language that the last name Sylvester comes from, used in the era of Rotrude's father, later known as?
**Answer:** Medieval Latin

**Turn 1:** `<think>` To answer this question, I need to find out which language the last name Sylvester comes from. Let's start by searching Wikipedia for this information. `</think>`
`<tool_call>` {"name": "wikipedia_search", "arguments": {"query": "Last name Sylvester comes from what language"}} `</tool_call>`
`<tool_response>` "Sylvester (singer)", Sylvester (singer) Sylvester James Jr. (September 6, 1947December 16, 1988), who used the stage name of Sylvester, was an American singer-songwriter. Primarily active in ... ... `</tool_response>`
`<memory>` The last name Sylvester comes from the Latin language. `</memory>`

**Turn 2:** `<think>` The memory mentions that the last name Sylvester comes from the Latin language. However, it does not provide any information about the form of the Latin language used in the era of Rotrude's father, later known as. I need to find out which era Rotrude's father lived in. `</think>`
`<tool_call>` {"name": "wikipedia_search", "arguments": {"query": "Who was Rotrude's father, and what was he later known as?"}} `</tool_call>`
`<tool_response>` "Rotrude", Rotrude Rotrude (or sometimes referred to as Hruodrud/Hruodhaid) (775/778 – 6 June 810) was a Frankish princess, the second daughter of Charlemagne from his marriage to Hildegard. Few clear records ... ... `</tool_response>`
`<memory>` The last name Sylvester comes from the Latin language. Rotrude was a Frankish princess, the second daughter of Charlemagne from his marriage to Hildegard. `</memory>`

**Turn 3:** `<think>` The memory mentions that the last name Sylvester comes from the Latin language. However, it does not provide any information about the form of the Latin language used in the era of Rotrude's father, later known as. I need to find out who Charlemagne was. `</think>`
`<tool_call>` {"name": "wikipedia_search", "arguments": {"query": "Who was Charlemagne"}} `</tool_call>`
`<tool_response>` "Charlemagne", Charlemagne Charlemagne or Charles the Great (2 April 742 – 28 January 814), numbered Charles I, was King of the Franks from 768, King of the Lombards from 774, and Holy Roman Emperor from ... ... `</tool_response>`
`<memory>` The last name Sylvester comes from the Latin language. Rotrude was a Frankish princess, the second daughter of Charlemagne from his marriage to Hildegard. Charlemagne was later known as Holy Roman Emperor. `</memory>`

**Turn 4:** `<think>` The last name Sylvester comes from the Latin language. Charlemagne was later known as Holy Roman Emperor. The question asks what was the form of the Latin language used in the era of Rotrude's father, later known as. Since Rotrude's father was Charlemagne, and Charlemagne was later known as Holy Roman Emperor, we need to find out what was the form of the Latin language used in his era. I will use the wikipedia_search function to find this information. `</think>`
`<tool_call>` {"name": "wikipedia_search", "arguments": {"query": "Form of Latin language used in the era of Charlemagne"}} `</tool_call>`
`<tool_response>` "Medieval Latin", Medieval Latin Medieval Latin was the form of Latin used in Roman Catholic Western Europe during the Middle Ages. In this region it served as the primary written language, though local languages were also written to ... ... `</tool_response>`
`<memory>` The last name Sylvester comes from the Latin language. Rotrude was a Frankish princess, the second daughter of Charlemagne from his marriage to Hildegard. Charlemagne was later known as Holy Roman Emperor. The form of Latin language used in the era of Charlemagne was Medieval Latin. `</memory>`

**Turn 5:** `<think>` The memory mentions that the form of Latin language used in the era of Charlemagne was Medieval Latin. `</think>`
The answer to the question is \boxed{Medieval Latin}.

