# OpenReview forum: "MemSearcher: Training LLMs to Reason, Search and Manage Memory via End-to-End Reinforcement Learning"
_ICLR.cc/2026/Conference — ICLR 2026 Conference Withdrawn Submission_

### Official Review · Reviewer_FP86 · 2025-10-19

**Soundness:** 3
**Presentation:** 3
**Contribution:** 2
**Rating:** 4
**Confidence:** 4

**Summary:**

MemSearcher introduces a memory-centric search agent that replaces ReAct’s ever-growing history with a compact, iteratively updated memory, stabilizing context length while preserving key facts. Trained via multi-context GRPO, it jointly optimizes reasoning, search, and memory management.

**Strengths:**

1. The paper presents a well-structured narrative from problem to method to results. Concepts (memory, actions, GRPO) are clearly defined.

2. The comparisons span diverse benchmarks (NQ, TriviaQA, HotpotQA, 2Wiki, Musique, PopQA, Bamboogle), covering in-/out-of-domain settings, model sizes (3B/7B), and strong baselines (RAG, ReAct variants, RL agents).

**Weaknesses:**

1. The paper performs end-to-end training of three capabilities: search engine calling, memory construction, and answer generation. However, the reported gains are joint, which obscures each component’s contribution. The work lacks ablations that isolate improvements from search, memory, and reasoning individually.

2. In Section 4.3.1, the discussion of whether RL is needed only compares against a no-training setup. A stronger baseline should include SFT, which was the prevailing approach before GRPO and is more informative than a no-training comparison.

3. The paper lacks thorough case analyses, with only one example in the appendix, especially for error and robustness studies. For example, the impact of incorrect memory updates and failure modes such as cyclic search queries that induce looping m t a o cycles are not analyzed.

**Questions:**

1. Beyond reducing context length, does the introduction of memory provide advantages for constructing search queries and answering questions compared to using the full LLM-Agent interaction history?

2. Since ReAct is training-free by default, would a GRPO-trained ReAct variant achieve similar improvements as those observed with your method?

3. Can the proposed method be applied to other base models such as Llama?

---

### Official Review · Reviewer_TvJF · 2025-10-26

**Soundness:** 2
**Presentation:** 3
**Contribution:** 2
**Rating:** 2
**Confidence:** 3

**Summary:**

They propose MemSearcher, an agent workflow that iteratively maintains a compact memory and combines the current turn with it.  To optimize this workflow, they introduce multi-context GRPO, an end-to-end RL framework that jointly optimize reasoning, search strategies, and memory management of MemSearcher Agents.

**Strengths:**

- The algorithm is clearly represented.
- The empirical results show that a 3B model trained with the algorithm has comparable performance to a 7B model.

**Weaknesses:**

- The novelty is limited: let the LLM to make a compact summary of the history intsead of directly using the whole history as input, using GRPO to fine-tune the agent, while masking the search engine output.
- The evaluation is not sufficient -- only one baseline ReSearch is used.

**Questions:**

See weakness.

---

### Official Review · Reviewer_ZTKw · 2025-10-26

**Soundness:** 2
**Presentation:** 3
**Contribution:** 2
**Rating:** 2
**Confidence:** 3

**Summary:**

This paper proposes MemSearcher, an LLM-based search agent that maintains a compact, iteratively updated memory instead of concatenating the full interaction history, aiming to improve efficiency and scalability in multi-turn reasoning. The authors further introduce multi-context GRPO, an extension of GRPO, to jointly train reasoning, search, and memory management in an end-to-end reinforcement learning framework. Experiments on seven public QA benchmarks show consistent improvements over ReAct-based and RL-trained search agents, with stable context length and reduced GPU memory usage. Overall, the paper addresses the growing context problem in LLM agents through a lightweight memory mechanism and an extended RL training algorithm.

**Strengths:**

1. The paper tackles an important issue in multi-turn LLM agents: context length explosion. It proposes a workflow that maintains a compact, iteratively updated memory.
2. The proposed multi-context GRPO is a natural and interesting extension of GRPO, offering a way to optimize reasoning, search, and memory jointly.
3. Experiments are conducted on several public benchmarks with reasonably clear comparisons, and the approach shows consistent improvement over baselines under the same training setup.

**Weaknesses:**

1. **Limited contribution to future LLM agent development.**
My main concern is that the claimed contribution to the broader development of LLM agents is limited. With modern large models such as GPT-5 already supporting very long context windows, the improvement demonstrated in the paper (via explicit context management) may largely stem from the weakness of smaller backbones (3B/7B). The results is limited to academic setting. It is unclear whether the proposed method will remain beneficial when scaled to stronger models.
I would be very interested to see experiments varying the model size and the trajectory length the agent must handle. For example: current search tasks involve roughly <10k tokens of context. how large must a model be to handle such trajectories effectively without explicit memory compression?
2. **Overly simple “memory” mechanism.**
The current memory update design is relatively naive and may only scale to the 10k-token regime. When the required context exceeds 100k+ tokens, more sophisticated mechanisms such as retrieval-augmented or structured memory (see RAG and related works) become necessary.
Specific issues include:
(a) Error accumulation: if critical information is omitted in one turn, it is permanently lost.
(b) Planning horizon bias: some information is only useful for long-term planning, but the current mechanism may discard it prematurely.
3. **Lack of statistical rigor.**
Experimental results do not report error bars or standard deviations. The observed improvements could be within natural training fluctuations.
4. **Writing and structure.**
The paper could be more concise. Much of the Method section reads like background or preliminary material and could be significantly shortened without loss of clarity.

**Questions:**

1. In your formulation, each turn is treated as an individual training sample, whereas most prior work regards the entire trajectory as a single training instance. How do you implement this?
2. Multi-turn GRPO requires a larger number of samples to estimate a reliable baseline, especially since each turn is treated as an independent training sample. Have you made any algorithmic modifications or variance-reduction techniques to address potential instability in baseline estimation?
3. The summarization (memory update) step introduces extra computational overhead per turn. Does this affect the overall training efficiency, and how significant is the slowdown compared with standard GRPO training?

---

### Official Review · Reviewer_L6Aq · 2025-10-31

**Soundness:** 3
**Presentation:** 2
**Contribution:** 2
**Rating:** 2
**Confidence:** 4

**Summary:**

The paper introduces Mem-search, a method that combines memory management (by having a context window) and reasoning to perform better for QA tasks. End-to-end RL is done to train memory management and reasoning directly on the model.

The paper ran experiments on various QA tasks and compared their methods with other methods, including vanilla CoT and other RL methods.

**Strengths:**

The paper's idea of searching and maintaining a memory bank is not novel, but training it end-to-end with RL is. The paper shows comprehensive results in some QA tasks that their method produces better-performing LLMs.

**Weaknesses:**

I have several major concerns about the quality and contribution of the work.

1. The contribution is incremental where we are merely doing end-to-end RL with a memory bank. While it shows performance improvement, not much design justification and ablation is done w.r.t. the results. For instance, what happens if we increase the memory bank size? Does performance improve (we expect so)? What is the difference in LLM response with and without the memory bank?

2. Do we really need a memory searcher for QA tasks? Let me propose a simple baseline: what happens if we simply do supervised finetuning over the LLM (PEFT, or full model) for the QA dataset and evaluate the performance? If I recall, in papers where we perform finetuning over certain QA tasks, we can actually get quite high performance. For instance, >70% for TriviaQA.

3. I'd appreciate some empirical design justification in maintaining a compact memory bank. For instance, why not show us what responses are possible with and without a memory bank? Why is having a memory bank important? What kind of question can we not tackle without a memory bank for certain QA tasks? Can we see examples of failed cases? Without having these insights, this paper becomes very incremental - just applying a RL-technique to train a new component (in this case, a compact memory). It's hard to justify accepting such papers.

3. The use of exact match metric is kind of strange - could you elaborate? Usually we do keyword extraction for better evaluation.

4. The smoothing technique in figure 5 to display the result seems weird. Why is the smoothed curve way below the original plot at the beginning?

**Questions:**

See weaknesses above.

---

### Note · Authors · 2026-01-06

I have read and agree with the venue's withdrawal policy on behalf of myself and my co-authors.